# Psychosocial Factors Associated with Lower Urinary Tract Symptoms One Year Postpartum

**DOI:** 10.3390/ijerph21010040

**Published:** 2023-12-27

**Authors:** Shayna D. Cunningham, Rogie Royce Carandang, Lisa M. Boyd, Jessica B. Lewis, Jeannette R. Ickovics, Leslie M. Rickey

**Affiliations:** 1Department of Public Health Sciences, University of Connecticut School of Medicine, Farmington, CT 06030, USA; carandang@uchc.edu; 2Virginia Polytechnic Institute, State University, Blacksburg, VA 24061, USA; lisamboyd@vt.edu; 3Department of Internal Medicine, Yale School of Medicine, New Haven, CT 06510, USA; jessica.lewis@yale.edu; 4Department of Social and Behavioral Sciences, Yale School of Public Health, New Haven, CT 06510, USA; jeannette.ickovics@yale.edu; 5Departments of Urology and Obstetrics, Gynecology & Reproductive Services, Yale School of Medicine, New Haven, CT 06510, USA; leslie.rickey@yale.edu

**Keywords:** depressive symptoms, perceived stress, social support, lower urinary tract symptoms, postpartum

## Abstract

Pregnancy carries substantial risk for developing lower urinary tract symptoms (LUTSs), with potential lifelong impacts on bladder health. Little is known about modifiable psychosocial factors that may influence the risk of postpartum LUTSs. We examined associations between depressive symptoms, perceived stress, and postpartum LUTSs, and the moderating effects of perceived social support, using data from a cohort study of Expect With Me group prenatal care (n = 462). One year postpartum, 40.3% participants reported one or more LUTS. The most frequent LUTS was daytime frequency (22.3%), followed by urinary incontinence (19.5%), urgency (18.0%), nocturia (15.6%), and bladder pain (6.9%). Higher odds of any LUTS were associated with greater depressive symptoms (adjusted odds ratio (AOR) 1.08, 95% confidence interval (CI) 1.04–1.11) and perceived stress (AOR 1.12, 95% CI 1.04–1.19). Higher perceived social support was associated with lower odds of any LUTS (AOR 0.94, 95% CI 0.88–0.99). Perceived social support mitigated the adverse effects of depressive symptoms (interaction AOR 0.99, 95% CI 0.98–0.99) and perceived stress (interaction AOR 0.97, 95% CI 0.95–0.99) on experiencing any LUTS. Greater depressive symptoms and perceived stress may increase the likelihood of experiencing LUTSs after childbirth. Efforts to promote bladder health among postpartum patients should consider psychological factors and social support.

## 1. Introduction

Pregnancy carries substantial risk for developing lower urinary tract symptoms (LUTSs) and is an exposure in 80% of United States (US) women by age 44 [1], leading to potential lifelong impacts on bladder health. Up to 52% of birthing people experience urinary incontinence (UI) during pregnancy [2], and 11–35% report LUTSs one year after delivery [3,4]. Though a large body of evidence links clinical parameters (e.g., age, body mass index (BMI), delivery mode, parity) with postpartum LUTSs [5], outcome measures typically focus on a single diagnosis (most commonly UI) [6,7,8,9,10], and little is known about modifiable psychosocial factors that may increase or reduce risk of LUTSs during the perinatal period.

Several longitudinal studies among the general population and clinic-based samples of adult women have demonstrated an association between depressive symptoms and risk of LUTSs [11,12,13,14,15]. Depressive symptoms have also been associated with progression of existing LUTSs [16,17,18]. Similarly, evidence suggests LUTSs are prospectively associated with the development of affective disorders [11,12,19,20,21]. However, the relationship between postpartum depressive symptoms and LUTSs is unclear. One study showed that UI was positively associated with postpartum depression [22], while others have not demonstrated a definitive association [23,24].

Stress has been hypothesized as a risk factor for LUTSs, either directly or via affective disorders [25]. However, few studies have examined stress in relation to LUTSs, and none among pregnant or postpartum populations. In studies among adult female patients, elevated psychological stress levels have been associated with LUTSs [26,27]. Stress and depression affect as many as 53% and 25% of pregnant people, respectively [28,29], warranting more research on their associations with postpartum LUTSs.

Research has highlighted the importance of social support—a multidimensional construct encompassing emotional, instrumental, or informational assistance received from others—as a psychosocial mechanism that may protect against adverse health outcomes directly and indirectly as a buffer against stressful circumstances [30,31,32]. Only a few studies have examined associations between perceived social support and LUTSs among women [33,34,35], one of which found low levels of social support to be prospectively associated with more burdensome LUTSs [33]. Another reported an association between low social support and greater prevalence and frequency of UI in a cohort of women aged 40 to 55 years old [34].

The objective of this study was to examine associations between depressive symptoms, perceived stress, and postpartum LUTSs and the direct and moderating effects of perceived social support. We hypothesized that higher levels of social support would be associated with lower odds of experiencing LUTSs and may buffer the adverse effects of depressive symptoms and perceived stress on postpartum LUTSs.

## 2. Materials and Methods

### 2.1. Data Source

These analyses utilized data collected as part of a prospective cohort study examining a group model of prenatal care (Expect With Me) implemented in four clinical sites in Nashville, TN, and Detroit, MI, that served a diverse population at risk for adverse birth outcomes from 2014 through 2016 [36,37]. Study participants were recruited from week 14 of pregnancy and followed through one year postpartum. Eligibility criteria included: less than 24 weeks pregnant; lack of a severe medical problem; ability to speak English or Spanish; and willingness to participate in the study. Individuals with a severe medical concern that a healthcare provider determined required more intensive monitoring via individual care were excluded. Staff at each clinical site explained the study to eligible participants, answered questions, and obtained informed consent. Analyses for this study used data collected through online surveys during the second trimester of pregnancy and twelve-months postpartum. Participants were paid USD 20 for each completed survey. The Institutional Review Boards at Yale, Vanderbilt, and Wayne State Universities approved all study procedures. The cohort was limited to study participants who provided data on LUTSs one year postpartum, resulting in an analytic sample of 462 pregnant individuals. Compared to pregnant individuals included in this analytic sample, those excluded were more likely to be younger and Black (*p* < 0.05). There were no other significant sociodemographic differences.

### 2.2. Measures

Demographic data including age, race and ethnicity (non-Hispanic White, non-Hispanic Black, Hispanic, and Other), educational attainment (less than high school, high school graduate/GED, and some college/higher education), relationship status (single/separated/divorced and married/living with partner), and insurance status (private, public, and none) were collected during the second trimester of pregnancy. The delivery mode for the most recent pregnancy (vaginal versus cesarean section), post-pregnancy BMI (underweight, healthy weight, overweight, and obese), depressive symptoms, perceived stress, perceived social support, parity, and LUTSs were assessed one year postpartum.
Depressive symptoms: Depressive symptoms were measured using the Center for Epidemiologic Studies-Depression (CES-D) scale [38]. Consistent with prior studies of pregnant individuals, five items describing somatic symptoms often associated with pregnancy (e.g., poor appetite, poor sleep) were excluded [39,40]. Participants rated how often during the past week they experienced affective components of depressed mood (e.g., feelings of loneliness, failure, hopelessness) on a 4-point Likert scale, ranging from none of the time (0) to all of the time (3). Scores for each item were summed, with a higher summative score indicating greater depressive symptoms (α = 0.87).Perceived stress: An abridged, four-item version of the Perceived Stress Scale was used to measure the degree to which an individual appraised situations in their life as stressful [41]. Participants rated how often during the past month they felt or thought a certain way (e.g., unable to control important things, not confident about ability to handle personal problems) on a 5-point Likert scale, ranging from never (1) to very often (5). Item scores were summed, with higher scores indicating higher levels of stress (α = 0.79).Social support: Social support was assessed using an abridged, four-item version of the Multidimensional Scale of Perceived Social Support [42]. Participants rated their level of agreement about the availability of a support person (e.g., someone to talk to about my problems, someone who helps me make decisions) on a 5-point Likert scale, ranging from strongly disagree (1) to strongly agree (5). A summary score was calculated, with higher scores indicating greater perceived social support (α = 0.92).LUTSs: Items used to assess LUTSs were adapted from existing self-report measures of urinary frequency, urgency, nocturia, incontinence, and pain (Urogenital Distress Inventory, Overactive Bladder Questionnaire, Bristol Female Lower Urinary Tract Symptoms Questionnaire, King’s Health Questionnaire, O’Leary Sant Index) [43,44,45,46,47]. Participants responded ‘yes’ or ‘no’ to the following items assessing bladder symptoms: “I go to the bathroom too frequently during the day”; “I often have a strong or overwhelming urge to urinate”; “I go to the bathroom too frequently at night”; “I leak urine”; and “I have bladder pain”. Participants who responded ‘yes’ to at least one symptom were categorized as having ‘any LUTS’.

### 2.3. Data Analysis

Descriptive statistics were used to summarize participant characteristics. Multiple logistic regressions were run to examine the association between psychosocial factors and LUTSs (both single symptom and ‘any LUTS’) one year postpartum, controlling for participant characteristics (age, race/ethnicity, education, relationship status, insurance, post-pregnancy BMI, parity, delivery mode, and study site). Moderation effects of perceived social support between depressive symptoms, perceived stress, and LUTSs were tested using interaction terms. Significant interactions were probed by simple slopes tests using the “margins” command to generate simple slopes for exposure variables (depressive symptoms, perceived stress) at −1 standard deviation (SD), 0, and +1 SD on the centered perceived social support variable [48]. All analyses were conducted using Stata 17.0 (StataCorp, College Station, TX, USA). A *p*-value of less than 0.05 was considered statistically significant.

## 3. Results

### 3.1. Participant Characteristics

Table 1 shows participant characteristics. At baseline, participants had a mean age of 25.6 years, and 61.5% self-identified as non-Hispanic Black, 18.2% as non-Hispanic White, 16.5% as Hispanic and 3.9% as other. The majority had some college or higher education (46.5%), were single (including separated or divorced, 56.5%), had public insurance (68.4%), and had a vaginal delivery for their most recent pregnancy (72.9%). Fifty-two percent of participants were primiparous. Forty-three percent had a post-pregnancy BMI characterized as obese. One year postpartum, 40.3% reported having one or more LUTSs. The most frequent LUTS reported was frequent daytime urination (22.3%), followed by urinary incontinence (19.5%), urgency (18.0%), nocturia (15.6%), and bladder pain (6.9%).

### 3.2. Association of Psychosocial Factors with LUTSs Postpartum

Table 2 shows the multiple logistic regression model results for the association between psychosocial factors and LUTS type at 12 months postpartum. Participants who reported greater depressive symptoms had higher odds of having any LUTS (adjusted odds ratio (AOR) 1.08, 95% confidence interval (CI) 1.04–1.11), frequent daytime urination (AOR 1.05, 95% CI 1.02–1.08), urgency (AOR 1.05, 95% CI 1.01–1.09), nocturia (AOR 1.07, 95% CI 1.04–1.11), urinary incontinence (AOR 1.06, 95% CI 1.03–1.10), and bladder pain (AOR 1.05, 95% CI 1.00–1.10). Participants who reported greater perceived stress also had higher odds of having any LUTS (AOR 1.12, 95% CI 1.04–1.19), urgency (AOR 1.10, 95% CI 1.01–1.20), and urinary incontinence (AOR 1.14, 95% CI 1.04–1.23) compared to those who reported lower perceived stress. In contrast, higher perceived social support was associated with lower odds of having any LUTS (AOR 0.94, 95% CI 0.88–0.99), nocturia (AOR = 0.89, 95% CI 0.82–0.95), and urinary incontinence (AOR = 0.92, 95% CI 0.86–0.99).

### 3.3. Moderation Effects

Perceived social support moderated the associations between depressive symptoms (interaction AOR 0.99, 95% CI 0.98–0.99) and perceived stress (interaction AOR 0.97, 95% CI 0.95–0.99) and any LUTS. In particular, perceived social support was protective against the adverse effects of depressive symptoms (interaction AOR 0.98, CI 0.98–0.99) and perceived stress (AOR = 0.97, 95% CI 0.95–0.99) on nocturia. However, these findings may not reflect clinical significance due to small effect sizes.

Figure 1 shows the simple slopes test of the significant interaction terms. There was a positive predictive relationship between depressive symptoms and any LUTS and nocturia among those low in perceived social support. There was a negative predictive relationship among those high in perceived social support. This pattern with LUTSs, perceived stress, and social support interaction was similar.

## 4. Discussion

LUTSs are common one year after childbirth and are more common in participants with greater depressive symptoms and perceived stress scores. Additionally, there may be varying levels of effect for specific LUTSs (e.g., urgency vs. UI). Participants who perceived higher levels of social support had lower odds of experiencing LUTSs one year postpartum. Social support also played a protective role against the adverse effects of depressive symptoms and the perceived stress of having LUTSs.

A majority of the literature regarding postpartum LUTSs has focused on demographic and clinical factors associated with bladder health. The current study took a broader social-biologic-ecologic approach, as proposed by the Prevention of Lower Urinary Tract Symptom Research Consortium in its assessment of potentially modifiable psychosocial factors [49]. The positive associations between depressive symptoms and perceived stress scores and experience of postpartum LUTSs in the current sample are consistent with those reported among non-pregnant populations [50,51]. For instance, among women in the COBaLT study [52], depression was associated with SUI and the sensation of incomplete emptying. Although a systematic review noted a positive association between somatization, depression, and anxiety at baseline and a LUTS or its progression during follow-up, the evidence remains insufficient for establishing a bidirectional association, causality, or longitudinal changes [53]. Notably, existing longitudinal studies have predominantly centered on men, warranting further research to clarify these relationships in women [54,55,56].

Our findings are also consistent with the notion that supportive relationships have the potential to positively influence health [30,31,32]. This may occur through psychobiological mechanisms such as the attenuation of the sympathetic–adreno–medullar, hypothalamic–pituitary–adrenocortical, and inflammatory stress responses [30,57,58]. The mechanisms are not well understood; however, common pathways involving cortisol release and corticotropin-releasing factor have been suggested by clinical and animal studies to link increased psychological stress and LUTSs [57,59]. It is also possible that supportive relationships may lead to early treatment seeking and, in turn, less severe or quicker resolution of LUTSs [33].

Rates of postpartum UI, particularly stress urinary incontinence (SUI), have been extensively studied, although there is significant heterogeneity in populations, length of follow-up, and UI definition used. There is less literature regarding other LUTSs in the postpartum period, such as urgency, frequency, and nocturia. Thus, we sought to more broadly characterize LUTSs. A 2019 study that utilized the ICIQ-FLUTS questionnaire assessed a range of storage and voiding LUTSs and, similar to our study, reported an approximately 40% rate of storage LUTSs, with urgency and nocturia being the most common [60].

Several limitations of this study must be acknowledged. At the time of data collection, there were no validated tools that included the range of storage symptoms we wanted to assess within a limited number of items. Although the LUTS items used have not been validated as a group, the individual items use language common in many validated tools or utilize International Continence Society (ICS) terminology. Additionally, LUTS measures were not administered during pregnancy, precluding our ability to examine incident LUTSs or changes in LUTSs over time. Future research should examine the impact of depressive symptoms on LUTSs across the perinatal period. This study also did not account for prior or current treatment of LUTSs or depressive symptoms and data were not available regarding comorbid conditions or medications that individuals may have been taking that might influence LUTSs. All measures used were subject to self-report bias. Participants were all enrolled in group rather than traditional individual prenatal care; thus, the sample may not represent all pregnant people at risk for postpartum LUTSs.

Strengths of this study include its assessment of multiple LUTSs among a diverse sample. This is the first study among birthing people that examined the effect of psychosocial variables on postpartum LUTSs risk and investigated the moderating role of perceived social support. The use of validated measures of depressive symptoms, perceived stress, and social support also strengthen the internal validity of our findings.

## 5. Conclusions

Pregnancy and childbirth represent a critical timepoint for the development of pelvic floor disorders that can persist throughout a person’s lifetime. Although the risk of postpartum LUTSs has traditionally been attributed to clinical factors, the current analysis demonstrates the importance of psychosocial factors as well. In light of the high prevalence of postpartum LUTSs and the attendant impact on the lives of individuals who give birth, future postpartum research should incorporate psychosocial factors in addition to individual biologic and behavioral contributors. Delineating these mechanisms will enable improved screening across the perinatal period and inform strategies to maintain postpartum bladder health and prevent LUTSs over the life course.

## Figures and Tables

**Figure 1 ijerph-21-00040-f001:**
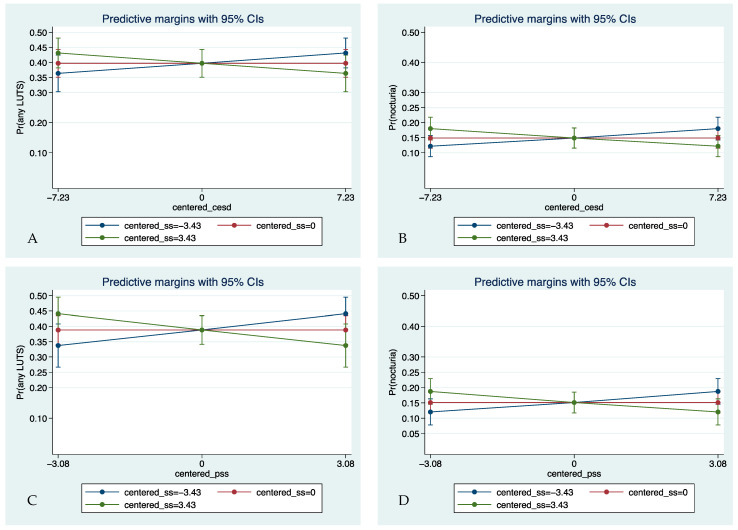
Interaction plots: (**A**) depressive symptoms × social support at any LUTS; (**B**) depressive symptoms × social support at nocturia; (**C**) perceived stress × social support at any LUTS; (**D**) perceived stress × social support at nocturia. Abbreviations: cesd, depressive symptoms; LUTS, lower urinary tract symptom; pss, perceived stress; ss, social support.

**Table 1 ijerph-21-00040-t001:** Participant characteristics (*n* = 462).

Characteristics	*n*	%
Demographics		
Age, mean (SD); range: 14–42 years	25.55 (5.55)	
Race and Ethnicity		
White, Non-Hispanic	84	18.2
Black, Non-Hispanic	284	61.5
Hispanic	76	16.5
Other, Non-Hispanic	18	3.9
Education		
Less than high school	64	13.9
High school graduate/GED	179	38.7
Some college or higher education	215	46.5
Relationship status		
Single or other (i.e., separated/divorced)	261	56.5
Married or living with partner	194	42.0
Insurance		
Private	134	29.0
Public	316	68.4
None	10	2.2
Post-pregnancy BMI group		
Underweight	9	1.9
Healthy weight	126	27.3
Overweight	127	27.5
Obese	200	43.3
Primiparous	239	51.7
Delivery mode of most recent child		
Vaginal	337	72.9
C-section	121	26.2
LUTSs variables (one year postpartum)		
Any LUTS currently	186	40.3
Frequent daytime urination	103	22.3
Urgency	83	18.0
Nocturia	72	15.6
Urinary incontinence	90	19.5
Pain	32	6.9
Psychosocial variables (one year postpartum)		
Depressive symptoms, mean (SD); range: 0–45	7.13 (7.23)	
Perceived stress, mean (SD); range: 4–18	8.33 (3.08)	
Perceived social support, mean (SD); range: 4–20	17.18 (3.43)	

GED—General Educational Development; BMI—body mass index; LUTSs—lower urinary tract symptoms; SD—standard deviation; percentage may not total 100% due to missing values.

**Table 2 ijerph-21-00040-t002:** Multiple logistic regression models for LUTSs postpartum.

Outcome						
Psychosocial Variables	Any LUTS	Frequent Daytime Urination	Urgency	Nocturia	Urinary Incontinence	Pain
	AOR (95% CI)	AOR (95% CI)	AOR (95% CI)	AOR (95% CI)	AOR (95% CI)	AOR (95% CI)
Depressive symptoms	1.08 (1.04, 1.11) ***	1.05 (1.02, 1.08) **	1.05 (1.01, 1.09) **	1.07 (1.04, 1.11) ***	1.06 (1.03, 1.10) ***	1.05 (1.00, 1.10) *
Perceived stress	1.12 (1.04, 1.19) **	1.07 (0.99, 1.15)	1.10 (1.01, 1.20) *	1.04 (0.95, 1.15)	1.14 (1.04, 1.23) **	1.03 (0.91, 1.17)
Perceived social support	0.94 (0.88, 0.99) *	0.96 (0.91, 1.03)	0.94 (0.87, 1.00)	0.89 (0.82, 0.95) **	0.92 (0.86, 0.99) *	0.93 (0.84, 1.02)
Depressive symptoms × Perceived social support	0.99 (0.98, 0.99) *	0.99 (0.99, 1.00)	0.99 (0.98, 1.00)	0.98 (0.98, 0.99) ***	0.99 (0.99, 1.00)	0.99 (0.99, 1.00)
Perceived stress × Perceived social support	0.97 (0.95, 0.99) *	0.99 (0.97, 1.02)	0.97 (0.95, 1.00)	0.97 (0.95, 0.99) *	0.98 (0.95, 1.00)	0.98 (0.95, 1.01)

LUTS—lower urinary tract symptom; AOR—adjusted odds ratio; CI—confidence interval. Statistical significance indicated by * *p* < 0.05; ** *p* < 0.01; *** *p* < 0.001; All models were adjusted for age, race and ethnicity, education, relationship status, insurance, post-pregnancy BMI, parity, delivery mode of most recent child, and study site.

## Data Availability

Data are available to qualified researchers upon reasonable request of the authors.

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
