# Peer review of "Psychosocial Factors Associated with Lower Urinary Tract Symptoms One Year Postpartum"

_ijerph, 2023, doi:10.3390/ijerph21010040_

Round 1

Reviewer 1 Report

Comments and Suggestions for Authors

It is a well written paper. Some suggestions:

1- It could be of interest to expand the pathophysiology of the association between depression or anxiety and LUTS in the discussion session (line 221-223) as for example the bidirectional relationship between depression and inflammatory disease states. 

2- The discussion lacks a broad evaluation of the results and it should be evaluated if there is any difference between the general population (Results of the COBaLT studdy - Zuluaga L doi: 10.1007/s00345-023-04351-w or the systematic review - Dina M. Mahjoob V doi.org/10.1016/j.cont.2023.100589) and the patients in pregnancy/postpartum.

Author Response

  1. It could be of interest to expand the pathophysiology of the association between depression or anxiety and LUTS in the discussion session (line 221-223) as for example the bidirectional relationship between depression and inflammatory disease states. 

We thank the reviewer for this suggestion and have added the following sentence to the Discussion section on Line 237-240:

“The mechanisms are not well understood, however common pathways involving cortisol release and corticotropin-releasing factor have been suggested by clinical and animal studies to link increased psychological stress and LUTS.”

  1. The discussion lacks a broad evaluation of the results and it should be evaluated if there is any difference between the general population (Results of the COBaLT studdy - Zuluaga L doi: 10.1007/s00345-023-04351-w or the systematic review - Dina M. Mahjoob V doi.org/10.1016/j.cont.2023.100589) and the patients in pregnancy/postpartum.

We have expanded the Discussion to include a description of how our study findings relate to research among the general population and other studies of pregnant and postpartum patients. We thank the reviewer for the references provided and have added these citations to the manuscript.

Reviewer 2 Report

Comments and Suggestions for Authors

The authors should be congratulated for their work. LUTS in the postpartum period and the psychological consequences are an intriguing topic, worthy to be published in this journal. However, several lacks should be addressed before considering any publication step:

- First, how does the LUTS were assessed? Was any validated questionnaire used? The authors should clarify this aspect.

- Any data on comorbidities of the cohort? LUTS could be associated with OSAS, circulatory disease, or diabetes (all these diseases are underexplored in the current manuscript highlighting a great lack of awareness and robustness of investigations before the study realization. Nocturia in OSAS patients should be explored due to its prevalence as well as urinary tract infections or diabetes. Moreover, the excessive daytime sleepiness of OSAS subjects (even if they did not have OSAS) could influence negatively the mental imbalance of patients who experienced that reality. The author should explore better these aspects

- Any data on the medications of patients? Diuretics and antihypertensive drugs could all influence the lower urinary tract symptoms. 

- The terms White and Black to treat Race/Ethnicity should be avoided. Despite the terms being deeply used in the previous literature, nowadays it is imperative to avoid discrimination in every form starting from scientific language

- Did pregnantants undergo ureteral stent placement during the pregnancy period? The ureteric stent placement is a very frequent event, due to positional changes of the uterus in pregnancy. Moreover, the presence of a stent or the removal in the last three months could influence the occurrence of LUTS (PMID= 38049673)

Author Response

The authors should be congratulated for their work. LUTS in the postpartum period and the psychological consequences are an intriguing topic, worthy to be published in this journal. However, several lacks should be addressed before considering any publication step:

  1. First, how does the LUTS were assessed? Was any validated questionnaire used? The authors should clarify this aspect.

Items used to assess LUTS were adapted from existing self-report measures of urinary frequency, urgency, nocturia, incontinence, and pain including the Urogenital Distress Inventory, Overactive Bladder Questionnaire, Bristol Female Lower Urinary Tract Symptoms Questionnaire, King’s Health Questionnaire, O’Leary Sant Index. We have added this detail to the Methods section on Lines 128-130.

In addition, limitations section on Lines 251-255 states the following:

“At the time of data collection, there were no validated tools that included the range of storage items we assessed within a limited number of items. While the LUTS items used are not validated as a group, the individual items use language common in many validated tools or utilize International Continence Society (ICS) terminology.”

  1. Any data on comorbidities of the cohort? LUTS could be associated with OSAS, circulatory disease, or diabetes (all these diseases are underexplored in the current manuscript highlighting a great lack of awareness and robustness of investigations before the study realization. Nocturia in OSAS patients should be explored due to its prevalence as well as urinary tract infections or diabetes. Moreover, the excessive daytime sleepiness of OSAS subjects (even if they did not have OSAS) could influence negatively the mental imbalance of patients who experienced that reality. The author should explore better these aspects.

Unfortunately, data on comorbidities are not available for this cohort. We have added text noting this in the paragraph describing the limitations of the study in the Discussion section on lines 259-261.

  1. Any data on the medications of patients? Diuretics and antihypertensive drugs could all influence the lower urinary tract symptoms. 

Data on medications were not collected for this cohort. We have added text noting this in the paragraph describing the limitations of the study in the Discussion section on lines 259-261.

  1. The terms White and Black to treat Race/Ethnicity should be avoided. Despite the terms being deeply used in the previous literature, nowadays it is imperative to avoid discrimination in every form starting from scientific language.

We agree that it is imperative to avoid discrimination in every form in scientific research. We have followed the National Institutes of Health style guide for describing race and national origin to describe our study population: https://www.nih.gov/nih-style-guide/race-national-origin#:~:text=NIH%20follows%20the%20Office%20of,Instead%2C%20indicate%20the%20specific%20groups. Our models include race and ethnic as covariates that represent social experiences, not biological facts, and have been careful that our interpretation of the results do not suggest otherwise.  

  1. Did pregnant patients undergo ureteral stent placement during the pregnancy period? The ureteric stent placement is a very frequent event, due to positional changes of the uterus in pregnancy. Moreover, the presence of a stent or the removal in the last three months could influence the occurrence of LUTS (PMID= 38049673)

This data is not available. This limitation is described in the Discussion section on line 258-259 as follows: “The study also did not account for prior or current treatment of LUTS…”.

Reviewer 3 Report

Comments and Suggestions for Authors

This study examined associations between depressive symptoms, perceived stress, and postpartum LUTS, and the moderating effects of perceived social support, using data from a cohort study of Expect With Me group prenatal care.

This is a well written manuscript. However, the data collection date back 20 2016. Please explain the delay in publication.

How did you determine the sample size.

The method section of sampling need more clarification.

How did you adjust the depressive symptoms during pregnancy with your results?

Which questionnaire was used to evaluate the LUTS? Please add the appropriate reference for this item.

Why did you choose Epidemiologic Studies-Depression scale for evaluation? since, Edinburgh Postnatal Depression Scale (EPDS) is developed for this purpose?

Please specify the study design in the title.

The discussion section need re-writing.

Comments on the Quality of English Language

Minor editing of English language required

Author Response

This study examined associations between depressive symptoms, perceived stress, and postpartum LUTS, and the moderating effects of perceived social support, using data from a cohort study of Expect With Me group prenatal care.

  1. This is a well written manuscript. However, the data collection date back to 2016. Please explain the delay in publication.

The primary aim of the Expect With Me study was examine the impact of this group prenatal care model on maternal health and birth outcomes. Additionally, the surveys used for the study yielded a wealth of data for which secondary analyses are still being conducted to address other important research questions. Several members of the research team also are investigators on the NIH-funded (NIDDK) multi-site collaborative center on the Prevention of Lower Urinary Tract Symptoms in Women. We were able to add questions used in this paper to our ongoing pregnancy study to address priorities from the multi-site urology team in this latter collaboration.

  1. How did you determine the sample size.

The sample size for the parent study was determined based on having 80% power to detect differences in birth outcomes for babies born to individuals enrolled in group versus individual care. The LUTS variables were added to the 12-month postpartum survey after the study began. Our sample includes all the participants who were asked and completed these items. Given we are examining simple associations, there is still sufficient power to detect small to moderate differences in the LUTS outcomes of interest among the subsample of participants from whom this data were collected.

  1. The method section of sampling need more clarification.

We have provided more detail in the Materials and Methods section starting on Line 76 on how sampling was conducted for the for the parent study. In addition, Lines 87-89 specify: “The cohort is limited to study participants who provided data on LUTS one year post-partum, resulting in an analytic sample of 462 pregnant individuals.”

  1. How did you adjust the depressive symptoms during pregnancy with your results?

This is cross-sectional study in which depressive symptoms are the exposure variable. Thus, we did not adjust for prenatal symptoms. Future research should examine the impact of depressive symptoms on LUTS across the perinatal period. We added a comment about this in the discussion, on lines 257-258.

  1. Which questionnaire was used to evaluate the LUTS? Please add the appropriate reference for this item.

Items used to assess LUTS were adapted from existing self-report measures of urinary frequency, urgency, nocturia, incontinence, and pain including the Urogenital Distress Inventory, Overactive Bladder Questionnaire, Bristol Female Lower Urinary Tract Symptoms Questionnaire, King’s Health Questionnaire, O’Leary Sant Index. We have added this detail, including appropriate references, to the Methods section on Lines 128-130.

  1. Why did you choose Epidemiologic Studies-Depression scale for evaluation? since, Edinburgh Postnatal Depression Scale (EPDS) is developed for this purpose?

Both the Center for Epidemiologic Studies Depression Scale (CESD) and Edinburgh Postnatal Depression Scale (EPDS) are both reliable instruments to assess depression during pregnancy (Heller et al., 2022). We chose the CESD because it has been widely used among pregnant and postpartum populations, including in previous studies conducted by our research team, in the United States context.

  1. Please specify the study design in the title.

We have clarified in the title that data were collected at one year postpartum to indicate the cross-sectional nature of the analysis. 

  1. The discussion section need re-writing.

We have expanded and re-organized the Discussion section.

Round 2

Reviewer 1 Report

Comments and Suggestions for Authors

Excellent work, there is no need for further corrections

Reviewer 2 Report

Comments and Suggestions for Authors

The authors reviewed the manuscript properly, answering all my questions.

Reviewer 3 Report

Comments and Suggestions for Authors

My comments have been responded satisfactorily.

Comments on the Quality of English Language

 Minor editing of English language required